# Piroplasm Infection in Domestic Cats in the Mountainous Region of Rio de Janeiro, Brazil

**DOI:** 10.3390/pathogens11080900

**Published:** 2022-08-11

**Authors:** João Pedro Palmer, Gilberto Gazêta, Marcos André, Aline Coelho, Laís Corrêa, José Damasceno, Carolina Israel, Rafael Pereira, Alynne Barbosa

**Affiliations:** 1Department of Microbiology and Parasitology, Biomedical Institute, Fluminense Federal University, Niterói, Rio de Janeiro 24210-130, Brazil; 2National Reference Laboratory for Rickettsiae Vectors, Oswaldo Cruz Institute, Oswaldo Cruz Foudation, Rio de Janeiro 21040-360, Brazil; 3Department of Pathology, Reproduction and Single Health, Júlio de Mesquita Filho College of Agrarian and Veterinary Sciences, Paulista State University, Jaboticabal 14884-900, Brazil; 4Veterinary Diagnostic Center, Niterói, Rio de Janeiro 24230-253, Brazil; 5Clinic School of Veterinary Medicine Luiz Cataldi de Souza, Serra dos Órgãos University Center, Teresópolis, Rio de Janeiro 25976-345, Brazil; 6Laboratory of Toxoplasmosis and Other Protozooses, Instituto Oswaldo Cruz, Fundação Oswaldo Cruz, Rio de Janeiro 21040-900, Brazil

**Keywords:** piroplasms, cats, PCR, Rio de Janeiro, Brazil

## Abstract

Piroplasm infections in domestic cats have been reported with increasing frequency in numerous countries. However, in some states of Brazil, little information is available about the occurrence of these parasites. Blood samples were collected from 250 cats treated at a private clinic in the mountainous region of Rio de Janeiro. The samples were each subjected to a blood count, microscopic examination, and molecular research on piroplasms. The animals’ clinical histories and epidemiological information were analyzed to identify the risk factors associated with infection. Ticks were recovered during the clinical care and were identified and subjected to molecular analyses to determine the presence of piroplasm DNA. Piroplasms were detected in 2.7% (7/250) of the cats. Nucleotide sequences of *Babesia vogeli* were identified in six cats, while the *Cytauxzoon* sp. was identified in one cat. Cats displaying apathy/weakness/prostration and hemorrhage/bleeding were more likely to be infected. In addition, *Amblyomma aureolatum* was recovered from a cat PCR-negative for piroplasms. This is the first study in Rio de Janeiro that has detected *Babesia vogeli* in cats. The results obtained here underscore the need for further studies in Rio de Janeiro to investigate the dynamics of such infections and the vectors involved.

## 1. Introduction

Vector-borne diseases comprise a variety of infections that can be caused by various pathogens, including protozoa such as those transmitted by ticks [1]. Ticks have a wide geographic distribution, and their occurrence can vary depending on the climatic and environmental conditions of each location as well as the population growth and mobility of different hosts, such as dogs and cats [1]. Piroplasms stand out among the protozoa transmitted by ticks in companion animals [2].

Piroplasm infections in felines can be determined mainly by observing the presence of protozoa of the genera *Babesia* and *Cytauxzoon*. In the former genus, *Babesia felis* and *Babesia leo* stand out and are mostly reported in symptomatic infections in cats in Africa and Asia [3,4]. In addition to these, other piroplasms, such as *Babesia gibsoni*, *Babesia vogeli*, *Babesia canis* and *Babesia microti*, which frequently infect other hosts, such as dogs and rodents, have also been reported infecting domestic cats in different countries, such as Saint Kitts and Nevis, Portugal, Italy, Thailand, Trinidad, China, Qatar and South Africa [1,5,6,7,8,9,10,11].

For feline babesiosis, the pathogenicity and clinical alterations vary according to the species of parasite involved, with anemia and associated clinical signs being predominant in the acute phase of the disease, including lethargy, anemia, fever, jaundice and neurological alterations. These are better characterized in cases of infections determined by observing the presence of *B. felis* [12,13].

*Cytauxzoon* spp. have a wide distribution, having been detected in cats in North and South America, Europe, Africa and Asia [14]. This genus includes *Cytauxzoon felis* and *Cytauxzoon manul*, species that infect domestic and wild cats in the United States and Europe, respectively [15,16,17,18,19]. In addition to these, *Cytauxzoon europaeus*, *Cytauxzoon banethi* and *Cytauxzoon otrantorum* were identified infecting wild cats in Europe [20]. Infections by *Cytauxzoon* sp. in domestic cats can occur asymptomatically or symptomatically, both with a wide variety of clinical signs. A feline with symptomatic cytauxzoonosis has nonspecific clinical signs, such as anorexia, lethargy and fever. Dyspnea, vomiting, jaundice, pale mucous membranes, tachycardia, generalized pain and vocalizations can also be observed [14]. Among the *Cytauxzoon* spp. that have been identified thus far, *C. felis* harbors the most concerns as it causes fatal diseases in cats, primarily in the USA [21].

In Brazil, piroplasms, such as *Babesia*/*Theileria* and *Cytauxzoon* sp., were detected by using molecular techniques in domestic cats in the states of Mato Grosso do Sul, São Paulo, Minas Gerais, Rio Grande do Sul, Rondônia and Distrito Federal [22,23,24,25,26], and *Cytauxzoon* sp. were detected by using molecular techniques in domestic cats of the states Rio Grande do Norte and Santa Catarina [27,28]. In Rio de Janeiro, the presence of intraerythrocyte inclusions compatible with piroplasms were detected in the blood of cats only by using microscopic parasitological techniques [29,30]. In this way, particularly in this state, there is a huge gap in epidemiological information about piroplasm infections among domestic cats. The aim of this study was therefore to reduce this information gap by analyzing the frequency, the hematological and clinical profiles of infections by these parasites as well as the risk factors inherent to infections in domestic cats in a city situated in the mountainous region of the state of Rio de Janeiro state.

## 2. Results

According to the information garnered from the forms, most of the cat owners, i.e., 74.8% (187/250), reported they lived in houses. In addition, 16.4% of the owners reported living in an apartment, 6.4% reported living on a farm, 0.8% reported that their cat did not have a fixed home but belonged to the local community and 1.6% did not report the type of housing. Most of the cats included in the study, i.e., 53.6% (134/250), were male, 57.6% (144/250) were more than one year old, 59.6% (149/250) were un-neutered and 95.6% (239/250) were of a mixed breed. As for health management, 62.4% (156/250) of the cats were treated preventively with endoparasiticides. However, 55.6% (139/250) of these animals had not received all the vaccinations recommended by veterinarians. Moreover, 70.4% (176/250) of the cat owners reported not administering ectoparasiticides, and 51.2% (128/250) did not habitually wash their cats. In addition, 54.4% (136/250) of the owners reported that their cats habitually scratched themselves, 1.2% (3/250) had a history of tick infestations and 67.6% (169/250) had a history of flea infestations. The questionnaire also revealed that most of the cats included in this study, i.e., 89.2% (223/250), had contact with other animals, particularly with domestic dogs (133/250); 57.6% (144/250) had access to a yard; 58.4% (146/250) had no access to a street and 66% (165/250) did not have access to vacant lots or woods. More than 80% (219/250) of the cat owners reported not having cat cages in their homes, and almost 50% (127/250) of the owners reported the presence of wild animals in the peridomicile, especially wild felids, such as jaguars (*Panthera onca*) and ocelots (*Leopardus pardalis*), which were reported in 6% (15/250) and 5.6% (14/250) of the cases, respectively.

In general, a molecular positivity rate of 2.8% (7/250) was found for piroplasms. Piroplasm merozoites and trophozoites were not found in the stained blood smears. The exploratory univariate analysis indicated that the variables showing statistically significant differences (*p* ≤ 0.2) in the frequency of infected cats were being neutered or spayed, having access to a yard, the presence of a cat cage at home, routine washing and frequent scratching. Among the piroplasm-positive cats, 5.1% (5/98) were neutered, 6.1% (6/99) did not have access to a yard and 2.3% (5/219) were not housed in cages in their owners’ home. In addition, 4.7% (6/128) of the piroplasm-positive cats were not washed, and 5.5% (6/109) of the piroplasm-positive cats did not habitually scratch themselves (Table 1).

Among the variables selected in the univariate exploratory analysis and used in the logistic regression model, only access to a yard showed a statistically significant difference (*p* ≤ 0.05) when compared with the frequency of infected cats. Cats with access to a yard were only 0.07 times more likely to be infected than cats without such access; hence, access to a yard was considered a protective factor since the odds ratio was less than one, being classified as a confounding factor (Table 2).

The only clinical and hematological changes evidenced in the positive cats were those described in Table 3. Among these changes, clinical signs, such as hemorrhage/bleeding, apathy/weakness/prostration and alterations in the platelet count, highlighting thrombocytopenia differed statistically in the univariate exploratory analysis (*p* ≤ 0.2). An analysis of the variables selected in the exploratory screening, together with the logistic regression model, revealed a statistically significant association with piroplasmid infections in cats that exhibited apathy/weakness/prostration and hemorrhaging/bleeding. Thus, cats with these alterations were more likely to be infected with these parasitic agents than cats without them (Table 4). Of the seven piroplasmid-positive cats, three were asymptomatic and four showed clinical changes, and of these, three showed changes compatible with piroplasmosis, including apathy, weakness and hemorrhage, as described in Table 4.

Of the seven samples positive for piroplasms detected by molecular analysis in the blood of the cats, six were discovered by using a nested PCR that amplified fragments of the ribosomal 18S RNA gene for *Babesia*/*Rangelia*/*Theileria*. All six showed nucleotide fragments with 99% to 100% identities with *Babesia vogeli* when compared with reference sequences deposited in GenBank from dogs and cats in Brazil and China. Of these sequences, five were at 493 bp at least, and the identification of *B. vogeli* was confirmed by considering the topography of the phylogenetic tree since the sequences obtained were inserted in the clade of this species with a high bootstrap value (Figure 1).

Furthermore, in a single sample, a nucleotide sequence compatible with *Cytauxzoon* sp. was obtained via the amplification of the 18S rRNA gene fragment detected by using another PCR protocol. This sequence showed a 99% identity match upon a comparison with *Cytauxzoon* sp. sequences from domestic cats and *Cytauxzoon* sp. of *Leopardus pardalis* (ocelots) from Brazil. In a comparison with a sequence of *Cytauxzoon manul* from Pallas’s cats (*Otocolobus manul*), an identity value of 95% was obtained. The molecular identification of the parasite was confirmed in the tree topology by means of the maximum likelihood method (Figure 2).

As for the molecular analysis of the cox-1 and hsp70 gene fragments, none of the blood samples from cats previously positive for piroplasms in the PCR assays based on the 18S rRNA gene tested positive in the additional PCR assays.

Among the four cats that presented some clinical alterations, four tested positive for *B. vogeli*. Furthermore, it was found that among the six cats that exhibited altered blood counts, five were positive for *B. vogeli* and one was positive for *Cytauxzoon* sp., highlighting the thrombocytopenia in cats positive for *B. vogeli*. Regarding epidemiological information, most of the owners reported that the cats testing positive for piroplasms did not have access to a street or a forest and reported that they had dogs at home (Table 5).

Only one female specimen of *Amblyomma aureolatum* was recovered from a cat. However, the DNA obtained from both the infested cat and the tick were negative in all the molecular assays (Figure 3).

## 3. Discussion

In this study, piroplasms were detected solely by using polymerase chain reactions of 2.8% of the blood samples from cats living in the mountainous region of Rio de Janeiro, with no evidence of parasitic intracellular inclusions detected via light microscopy. Using molecular techniques, frequencies higher than those of this study were reported in stray cats living in other Brazilian states, including Mato Grosso do Sul, Rondônia, São Paulo, Minas Gerais and Santa Catarina [22,23,24,28]. Such higher frequencies were also observed in blood samples from stray cats in the city of Rio de Janeiro using only the microscopic parasitological technique [29]. Piroplasm positivity rates higher than those of cats living in Teresópolis, RJ, were also reported in other countries, such as the United States, Saint Kitts and Nevis and Portugal [1,5,19]. However, lower frequencies were reported in free-ranging cats in Thailand and cats living in Spain and China [7,9,15]. The different frequencies reported in epidemiological studies may be directly attributed to the different genera of investigated and detected piroplasms, laboratory techniques, sample sizes, types of cat populations included, geographic locations and the ways the animals were handled.

The analysis of the possible risk factors associated with the infection of felines by piroplasms indicated that the owners’ statements about allowing their cats access to front or backyards ended up being misleading since the majority of the positive cats did not have access to backyards, according to the owners’ reports. It should be noted that this result must be analyzed with caution since the number of positive cats was low in relation to the sample number as a whole. Moreover, possible infections may be ascribed to a lack of knowledge about the previous histories of cats adopted by their owners as well as a proximity with other animals infested with ticks. Although questions about the origins of the animals were not included in the questionnaire, during conversations with owners during the clinical care, it was identified that all the positive cats originated from rescues. Factors intrinsic to owners cannot be ruled out either, such as a lack of knowledge about the usual behaviors of their own cats at different times of the day and even statements inconsistent with reality that cat owners make to avoid embarrassment.

In general, only a few cats exhibited hemorrhaging/bleeding or apathy/weakness/prostration. However, this small group included cats infected with piroplasms. This context ended up underscoring these clinical changes as being associated with infections by these parasitic agents. Changes involving lethargy and depression, which are similar to apathy, included as a clinical alteration caused by piroplasms in the cats of this study, were pointed out by other authors as possible signs of feline piroplasmosis [12,31]. Thus, further studies are needed that associate parasitology with clinical veterinary medicine comprising larger groups of domestic cats in order to truly consolidate these changes with parasitic infections.

In this study, piroplasms were detected in seven blood samples from cats only in the polymerase chain reactions through an analysis of the 18S rRNA gene fragment. Similar to in this study, DNA from piroplasms in blood samples obtained from cats did not amplify through molecular analyses with other gene targets, such as cox-1 and hsp70 [24]. Low parasite loads in the feline blood samples may have been one of the limiting factors for the non-amplification of DNA with the cox-1 and hsp70 targets, which are based on conventional PCR protocols with a single step of amplification, unlike the model of nested PCR.

In addition, among the seven samples with amplicons, six positive samples showed nucleotide sequences of *Babesia vogeli*. Unfortunately, comparisons with the literature at the national level were difficult because there are few studies that have investigated these agents in blood samples from cats since the literature is more focused on research involving dogs. Nevertheless, in Brazil, *B. vogeli* was identified through molecular analyses using as a target the 18S rRNA gene in blood samples from six stray cats that were roaming around the São Paulo Zoo as well as in those of eight stray cats and pet cats in Mato Grosso do Sul; two cats treated at the Veterinary Hospital in Rio Grande do Sul; two pet cats, one treated at a Zoonosis Control Center in Minas Gerais and one treated at a University in Rondônia and cats treated at private clinics in Brasília [22,23,24,25,26]. Therefore, this is the first study in which *B. vogeli* was detected in cats in Rio de Janeiro. *B. vogeli* was detected in domestic cats not only in Brazil but also in other countries, such as China, Saint Kitts and Nevis, Thailand, Qatar, Trinidad and Tobago and Portugal [1,5,7,8,9,10].

Although *B. vogeli* has been frequently detected in domestic dogs in several countries, including Brazil [2], this piroplasm is still rarely reported in the diagnosis of hemoparasites in samples from domestic cats [9]. It should be noted that cats have been gaining ground as companion animals and getting increasingly physically comfortable with dogs, which favors the exchange of pathogens, including those transmitted by ticks, particularly *Rhipicephalus sanguineus* (*sensu lato*), the red dog tick, which is distributed worldwide. Furthermore, it was observed that *B. vogeli* sequences from the blood of cats included in the present study showed high identities (99.7%) when compared with *B. vogeli* sequences from a dog sample also from the same region, reported in a previous study [32]. This panorama reinforces the possibility of the transmission of the parasites between dogs and cats in the mountainous region of Rio de Janeiro.

It is known that *B. vogeli* may cause various ailments in dogs ranging from asymptomatic infections to clinical forms of moderate pathogenicity [33]. However, the range of clinical changes this agent may cause in domestic cats is still unknown [5,9]. In Teresópolis, a state of Rio de Janeiro, Brazil, asymptomatic cats as well as cats with nonspecific clinical alterations tested positive for molecular diagnoses for *B. vogeli*. However, cats testing positive for *B. vogeli* were found to have hematological alterations compatible with piroplasmosis, particularly thrombocytopenia. Nevertheless, the possibility that these changes may be caused by other uninvestigated etiologies cannot be ruled out.

In general, piroplasm species typical of dogs in blood samples from cats have been sporadically detected by using molecular methods [9]. Sequencing-associated PCR is known to be essential for diagnosing feline piroplasmosis since it is considered the most sensitive and specific technique in the diagnosis of these parasites. PCR is able to detect infections with low circulating parasite loads, profiles usually revealed in blood samples from infected felines [12].

In this study, a specimen of *A. aureolatum* was collected from a cat during a clinical examination. However, DNA from the protozoa was not detected in either of the hosts. Despite the negative result, the discovery of this infestation confirms evidence of the possible transmission of *Rangelia vitalii* to cats. *Amblyomma aureolatum* was proven to be the only tick that transmits *R. vitalii* among canids [34]. In Rio de Janeiro, *A. aureolatum* is usually found in areas of preserved forest where crab-eating foxes (*Cerdocyon thous*), the main reservoir of the piroplasmid [35], exist.

In addition to these piroplasms, DNA from the *Cytauxzoon* sp. with a high identity with *Cytauxzoon felis* was detected in the blood of an un-neutered male cat, which had unimpeded access to a backyard, dense forest and other animals. The detection of this parasite was fortuitous since the cat appeared to be clinically healthy and the hematological diagnosis was simply a screening procedure performed prior to the animal being neutered. In general, the *Cytauxzoon* sp. has been reported in other cities and states in Brazil, based on molecular analyses of the blood of domestic cats in random cases [27] and in epidemiological surveys, at a frequency ranging from 0.6% to 41.9% [23,24,28], as found in the present study. In Brazil, fatal feline cytauxzoonosis was described in two lions housed at the Volta Redonda Zoo, in the state of Rio de Janeiro [36]. Wild felids in Brazil are considered its possible reservoirs [37]. This situation means that cities in the mountainous region of Rio de Janeiro, including Teresópolis, are favorable for the transmission of this hemoparasite between domestic and wild animals since there are areas with preserved Atlantic Forests located in three conservation units in this region. The fact that the cat with *Cytauxzoon* sp. had not been neutered and was circulating freely in different environments seems to have favored its infestation by arthropods and hence its infection.

As recommended in the literature, we decided on caution in the identification of the parasite at the taxonomic level of the genus *Cytauxzoon* due to a set of factors [24]. These factors included: a lack of knowledge about the tick’s vector in Brazil since there are no notifications of *Amblyomma americanum* and *Dermacentor variabilis* in this country, the nucleotide sequence analyzed here corresponds to only a small DNA fragment of the 18S rRNA gene and the fact that an infection evidenced in domestic cat in this country has an asymptomatic character, similar to that found in the positive cat included in the present study.

Although a feasible sample number was thought of, considering the previous frequencies reported in the literature, we know that the sampling that was used did not portray the different neighborhoods of the city nor even the different populations of cats. Thus, it is important to emphasize the need for further studies on piroplasmids in cat populations in order to epidemiologically enrich the literature, including a larger and more diversified sample panel in relation to the cat population composed of stray cats and those kept under the care of institutions dedicated to adoption.

## 4. Materials and Methods

### 4.1. Study Location and Sample Collection

The cats included in the study were treated at the veterinary clinic of private college located at a private university in the city of Teresópolis (latitude: −22.4123; longitude: −42.9664, 22°24′44″ S, 42°57′59″ W). This municipality lies in the mountainous region of the state of Rio de Janeiro at an altitude varying from 700 to 2263 m above sea level. The region has an average annual rainfall of 1721 mm and average annual temperature of 18.2 °C, varying from 3 °C in winter to 36.6 °C in summer, which characterizes its climate as oceanic temperate [38].

The biological samples were collected between September 2020 and August 2021, this being a cross-sectional study. Upon their arrival at reception desk of the veterinary clinic, the animal owners were informed about all the stages of the project and invited to participate in it. It should be noted that clinical care for cats occurred twice a week. This way, once a week, the owners with their cats were invited to participate in the study. There weren’t restrictions on including the animals. Those who accepted signed an informed consent form and filled out to a semi-structured questionnaire to provide general data about the cats as well as information regarding piroplasm infections. During the consultation, the animals were examined for the presence of arthropods, which were collected and then placed in 50 mL conical bottom tubes containing isopropanol. In addition, blood was collected from each animal’s saphenous or cephalic vein (approximately 1.5 mL of each animal) and stored in hematology tubes with EDTA.

Due to the scarcity of information on the frequency of piroplasmids in Rio de Janeiro obtained through molecular diagnosis, the confidence interval for the sample calculation was based on previous studies carried out in other states in Brazil. For *Babesia* sp., rates that ranged from 0.6% to 12% were taken into account [22,23,24,25,26], and for *Cytauxzoon*, rates that ranged from 0.6% to 41.9% were taken into account [23,24,26,28]. In this way, the blood collection of 240 domiciled cats was estimated considering a confidence interval of 97%.

### 4.2. Examination of Piroplasms under Light Microscopy

The presence of piroplasms was determined by examining two thin blood smear slides prepared from each sample using 5 µL of blood transferred to each microscope slide. One of the slides was stained with Giemsa solution (Merck^®^; Darmstadt, Germany) 1 drop of Giemsa diluted in 1 mL of phosphate buffer), and the other was subjected to rapid staining with a Newprov^®^ kit (Pinhais, Paraná, Brazil) as recommended by the manufacturer. All the slides were read under an Olympus BX 41 microscope at 1000× magnification.

### 4.3. Complete Blood Count (CBC)

Blood samples were subjected to hematological analysis in an automated device Icounter 3D (Diagno®), Belo Horizonte, Brazil obtain information, such as hematocrits, leukocyte counts and platelet counts. The reference values used for adults and kittens were, respectively, hematocrits (%) of 24–45 and 22–38; hemoglobinometry (g/dL) of 8–15 and 7–14; mean corpuscular volumes of 39–55 and 40–55; mean cell hemoglobin concentrations of 31–35 and 31–35; leukometry (mm^3^) of 5500–19,500 and 6000–17,000 and platelet counts (mm^3^) of 200,000–800,000 and 200,000–800,000 [39].

### 4.4. Taxonomic Identification of Ticks

The tick species were identified based on the taxonomic keys [40]. To this end, the specimens were placed in petri dishes with isopropanol and examined under a Dimex MZS-250 stereoscopic microscope. After identifying the taxa, lifecycle stages and sexes of adult ticks, the data were recorded in a registry book and the ticks were immediately placed individually in 1.5 mL microtubes and stored in a fridge. Photomicrographs were taken of the ticks using a Leica M205C stereoscopic microscope digital camera at 50 and 60× magnification and developed using Leica Application Suite v4.8 software. The identifications of the ticks were presented at the lowest possible taxonomic levels based on morphological analyses.

### 4.5. Molecular Analysis

All the blood samples were subjected to molecular analyses. DNA was extracted from the samples using the QIAamp^®^ DNA Blood Mini kit (Qiagen^®^, Hilden, Germany), as recommended by the manufacturer. After extraction, the eluted DNA was stored at −20 °C. Subsequently, it was subjected to a polymerase chain reaction (PCR) to detect gene fragments encoding GAPDH [41] to verify that DNA was properly extracted.

The material was examined with primers for *Babesia*, *Rangelia* and *Theileria* piroplasms using nested PCRs [42]. In addition, *Cytauxzoon* sp. DNA was analyzed using the protocol described by Birkenheuer [21]. All the PCRs were performed using Promega^®^, Madison, WI, USA master mix, and the final amplified product was visualized in 1.5% agarose gel stained with GelRed. This amplified DNA was purified using a Wizard^®^ SV Gel and PCR Clean-Up System (Promega^®^, Madison, WI, USA), as recommended by the manufacturer. Positive samples were also subjected to PCR assays based on the amplification of fragments of the genes cox1 [43] and hsp70 [44].

Tick DNA was obtained using the NaCl extraction technique [45] with adaptation, which consisted of using 360 µL of 5 M NaCl solution. After extraction, the DNA was quantified in a NanoDrop spectrophotometer (Thermo Fisher Scientific, Waltham, MA, USA). The primers described in [21,42], which were used in the molecular detection of these protozoa in blood samples obtained from cats, were also used in the search for piroplasms in tick DNA. PCR Master Mix (Promega^®^, Madison, WI, USA) was used in all the tick DNA amplifications, and the amplified products were visualized on a 1.5% agarose gel stained with GelRed.

In all DNA extractions and PCRs, negative controls were used that corresponded to ultrapure water, and positive controls (*Babesia vogeli* and *Cytauxzoon* sp.) were used from other projects of the team. Sequence trimming and consensus sequencing were performed using Lasergene SeqMan Madison, WI, USA software version 7.0 (DNAStar Inc.). The analytical tool BLASTN was then used to compare the data obtained against reference sequences belonging to the same gene fragment stored in the GenBank database. Consensus sequences were saved in FASTA format and aligned with other homologous sequences taken from the GenBank database using MEGA X software (https://www.megasoftware.net/, accessed on 20 January 2022). Phylogenetic inferences about the gene sequences that were used were obtained from maximum likelihood analyses for confirmation with a bootstrap accessed with 1000 replications, and the best evolutionary model that was selected was based on the Akaike information criterion (AIC) using the W-IQ-TREE software (http://iqtree.cibiv.univie.ac.at/, accessed on 2 February 2022). The phylogenetic tree was edited and rooted using MEGA X software. Note that the selected reference sequences followed the ten clades proposed [46].

### 4.6. Statistical Analysis of the Results

General information about the cats, such as each cat’s type of home, sex, age and breed, whether or not the animal was neutered, treated with an ectoparasiticide, had access to a street or woods, had contact with other animals, had a history of tick infestations and was routinely washed and whether there was the presence of wild animals in the peridomicile, among other things, was obtained from questionnaires the owners were asked to fill out. This information, as well as that pertaining to the CBC and that was garnered from the animals’ clinical examinations, was stored in Microsoft Office Excel 2007^®^ databases.

Positivity for piroplasms was demonstrated by means of descriptive frequencies, and a statistical analysis was performed to ascertain the presence or absence of a significant association between the information obtained from the questionnaire, blood count and clinical examination of each animal with piroplasm infection. This was done by performing a univariate exploratory analysis of the data to select the variables with *p* ≤ 0.2 using the chi-squared test (for variables with more than two categories) or Fisher’s exact test (for variables with only two categories). The significant variables were then subjected to a multivariate analysis using logistic regression at a 5% level of significance. The existence of factors associated with piroplasm infection was estimated by using odds ratio (OR) and its respective 95% confidence intervals. All the analyses were performed using the CDC’s Epi Info^TM^ software (https://www.cdc.gov/epiinfo/index.html, accessed on 20 January 2022).

## 5. Conclusions

The molecular analysis performed in this study revealed the presence of DNA from different piroplasms in the blood of pet cats treated at a veterinary clinic in the mountainous region of Rio de Janeiro, Brazil. The piroplasms, identified by analyzing the 18S rRNA gene nucleotide sequences, consisted mainly of *B. vogeli* and *Cytauxzoon* sp., being the first having already been detected in blood samples from dogs in this region. However, in cats, the results obtained here are pioneering findings. This reality underscores the need for veterinarians to be vigilant during the clinical examination of cats since these animals may actually be infected and presenting clinical signs of feline piroplasmosis, which is still little known.

## Figures and Tables

**Figure 1 pathogens-11-00900-f001:**
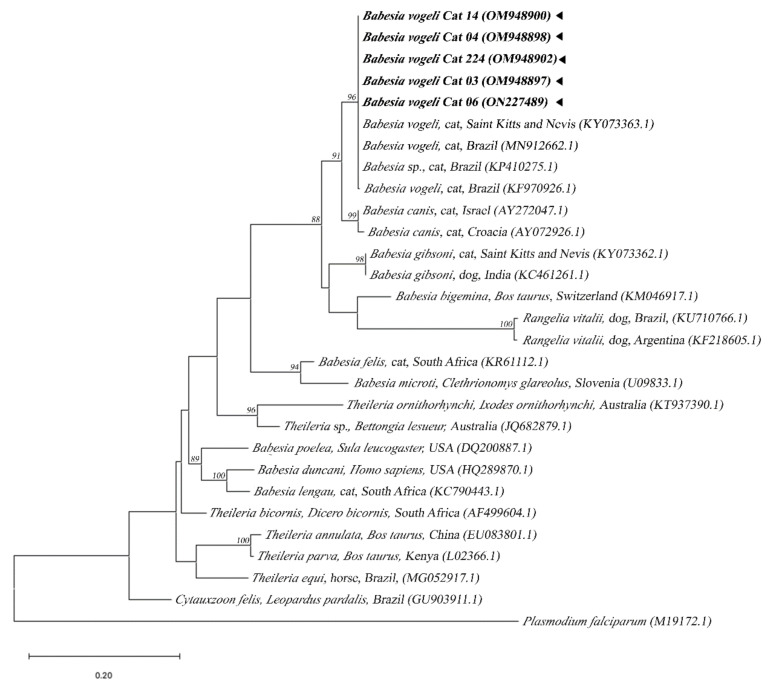
Phylogenetic tree based on alignment of 493 bp of the DNA fragment of 18S RNAr gene from piroplasms using the maximum likelihood method with a TIM2 + F + I + G4 evolutionary model. Sequences from this study are each highlighted in bold and with a triangle. *Plasmodium falciparum* DNA sequence was used as an outgroup.

**Figure 2 pathogens-11-00900-f002:**
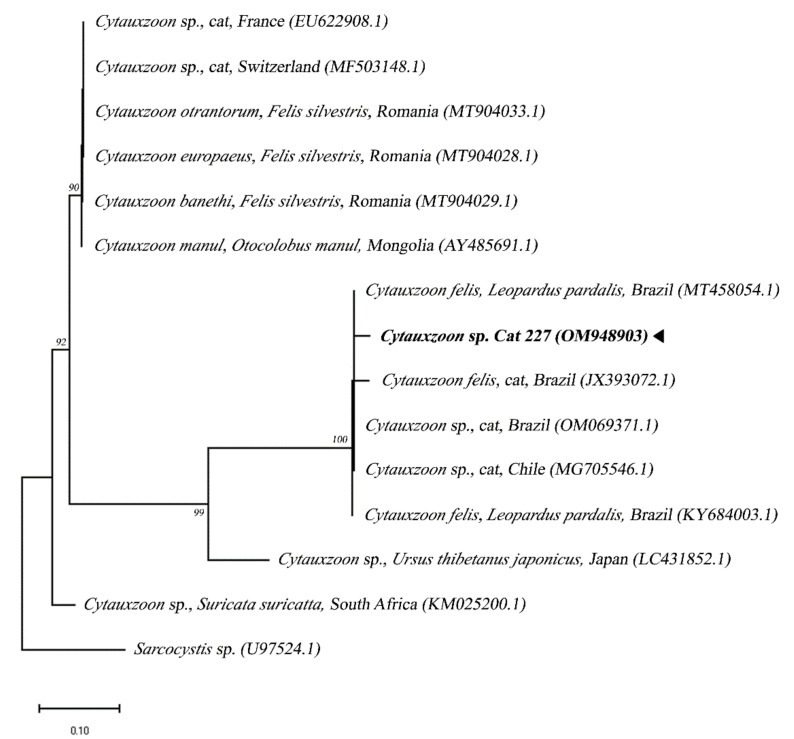
Phylogenetic tree based on the alignment of 230 bp of the DNA fragment of the 18S RNAr gene from *Cytauxzoon* sp. using the maximum likelihood method with the evolutionary model GTR + F + G4. Sequences of this study are each highlighted in bold and with a triangle. A sequence of *Sarcocystis* sp. was used as an outgroup.

**Figure 3 pathogens-11-00900-f003:**
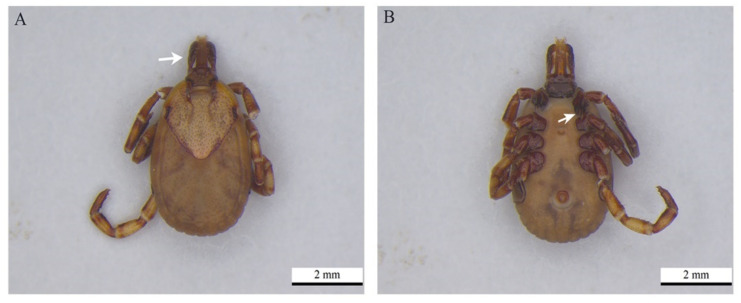
*Amblyomma aureolatum* female recovered from a cat during clinical care: (**A**) dorsal view, palps longer than wide being pointed by the arrow and (**B**) pair of spines on the first coxae on the first pair of legs at the arrowhead.

**Table 1 pathogens-11-00900-t001:** Frequency of blood samples positive and negative for piroplasms, according to information garnered from the questionnaire filled out by the owners of cats treated at clinic in Teresópolis, Rio de Janeiro, Brazil.

Variable			Positive		Negative		*p* Value
		Total	*n* = 7	%	*n* = 243	%	
Type of dwelling	House	187	5	0	182	97.3	1.159 ^a^
	Apartment	41	2	4.9	39	95.1	
	Farm	16	0	0	16	100	
	Community	2	0	0	2	100	
	Undetermined	4	0	0	4	100	
Sex	Male	134	3	2.2	131	97.8	0.707 ^b^
	Female	116	4	3.4	112	96.6	
Age	≤1 year	96	1	1.0	95	99	0.248 ^b^
	>1 year	144	6	4.2	138	95.8	
	Undetermined	10	0	0	10	100	
Breed	Mixed breed	239	7	2.9	232	97.1	1.000 ^b^
	Pure breed	11	0	0	11	100	
Neutered	Yes	98	5	5.1	93	94.9	0.117 ^b^ *
	No	149	2	1.3	147	98.7	
	Undetermined	3	0	0	3	100	
Use of anti-parasitic drugs	Yes	156	6	3.8	150	96.2	0.263 ^b^
	No	92	1	1.1	91	98.9	
	Undetermined	2	0	0	2	100	
Vaccination	Yes	76	2	2.6	74	97.4	0.615 ^b^
	No	139	2	1.4	137	98.6	
	Undetermined	35	3	8.6	32	91.4	
Use of anti-ectoparasitic drugs	Yes	72	2	2.8	70	97.2	1.000 ^b^
	No	176	5	2.8	171	97.2	
	Undetermined	2	0	0	2	100	
Yard access	Yes	144	1	0.7	143	99.3	0.19 ^b^ *
	No	99	6	6.1	93	93.9	
	Undetermined	7	0	0	7	100	
Cat cage at home	Yes	26	2	7.7	24	92.3	0.163 ^b^ *
	No	219	5	2.3	214	97.7	
	Undetermined	5	0	0	5	100	
Street access	Yes	102	1	1.0	101	99	0.245 ^b^
	No	146	6	4.1	140	95.9	
	Undetermined	2	0	0	2	100	
Access to woods	Yes	79	1	1.3	78	98.7	0.433 ^b^
	No	165	6	3.6	159	96.4	
	Undetermined	6	0	0	6	100	
Contact with other animals	Yes	223	7	3.1	216	96.9	1.000 ^b^
	No	24	0	0	24	100	
	Undetermined	3	0	0	3	100	
History of infestation	Ticks	3	0	0	3	100	1.112 ^a^
	Fleas	169	4	2.4	165	97.6	
	Ticks and fleas	8	0	0	8	100	
	No	67	3	4.5	64	95.5	
	Undetermined	3	0	0	3	100	
Routine washing	Yes	118	1	0.8	117	99.2	0.122 ^b^ *
	No	128	6	4.7	122	95.3	
	Undetermined	4	0	0	4	100	
Habit of scratching	Yes	136	1	0.7	135	99.3	0.046 ^b^ *
	No	109	6	5.5	103	94.5	
	Undetermined	5	0	0	5	100	
Presence of wild animals in the peridomicile	Yes	127	2	1.6	125	98.4	0.448 ^b^
	No	118	5	4.2	113	95.8	
	Undetermined	5	0	0	5	100	
Total		250	7	2.8	243	97.2	

^a^: Chi-squared test; ^b^: Fisher’s exact test; * *p* ≤ 0.2.

**Table 2 pathogens-11-00900-t002:** Final model of logistic regression analysis for epidemiological information concerning the detection of piroplasms in blood samples from cats treated at a private veterinary clinic in Teresópolis, Rio de Janeiro, Brazil.

Variable	Coefficient	Standard Error	*p* Wald Test	Degrees of Freedom	*p* Value	Odds Ratio (CI 95%)
Neutered	0.6122	0.948	0.6458	5	0.5184	0.5422
Yard access	2.5576	1.126	2.2714	5	0.023 *	0.0775
Cat cage at home	0.7781	0.9798	0.7941	5	0.4271	0.4593
Routine washing	2.0719	1.1655	1.7777	5	0.0755	7.9401
Habitual scratching	2.1168	1.1197	1.8905	5	0.0587	8.3041

* *p* ≤ 0.05.

**Table 3 pathogens-11-00900-t003:** Frequency of blood samples positive and negative for piroplasms, according to information obtained from the clinical changes compatible with piroplasmosis and the hematological analysis collected from cats treated at a private veterinary clinic in Teresópolis, Rio de Janeiro, Brazil.

Variable		Total	Samples Positive for Piroplasms		Samples Negative for Piroplasms		*p* Value
			*n* = 7	%	*n* = 243	%	
Hemorrhaging/bleeding	Yes	3	1	33.3	2	66.7	0.081 ^b^ *
	No	247	6	2.4	241	97.6	
Apathy/weakness/prostration	Yes	3	2	66.7	1	33.3	0.001 ^b^ *
	No	247	5	2	242	98	
Red blood cell count	Anemia	4	0	0	4	100	0.392 ^a^
	Normal	191	6	3.1	185	96.9	
	Erythrocytosis	55	1	1.8	54	98.2	
Leukocyt count	Leukocytosis	11	1	9.1	10	90.9	1.784 ^a^
	Normal	185	5	2.7	180	97.3	
	Leukopenia	54	1	1.9	53	98.1	
Platelet count	Thrombocytopenia	93	3	3.2	90	96.8	0.171 ^a^ *
	Normal	154	4	2.6	150	97.4	
	Thrombocytosis	3	0	0	3	100	
Total		250	7	2.8	243	97.2	

^a^: Chi-squared test; ^b^: Fisher’s exact test; * *p* ≤ 0.2.

**Table 4 pathogens-11-00900-t004:** Final model of logistic regression analysis for the clinical and hematological changes associated with the detection of piroplasms in blood samples from cats treated at a private veterinary clinic in Teresópolis, Rio de Janeiro, Brazil.

Variable	Coefficient	Standard Error	*p* Wald Test	Degrees of Freedom	*p* Value	Odds Ratio (CI 95%)
Apathy/Weakness/Prostration	4.7971	1.3264	3.6141	3	0.0003 *	121.1616
Hemorrhaging/Bleeding	3.347	1.3475	2.4838	3	0.013 *	28.4162
Platelet count	0.1843	0.8734	0.211	3	0.8329	0.8317

* *p* ≤ 0.05.

**Table 5 pathogens-11-00900-t005:** General epidemiological information, as well as clinical and hematological changes, found in piroplasmid-positive cats detected by using molecular analysis.

Animal Data	Epidemiology	Clinical Alteration	Hematological Alteration	Nested PCR	PCR	Sequencing	Sequence Size	Accession Number
Female, 2 years old, neutered	Lives in house	Vomiting	No hematological changes	Positive	Negative	*Babesia vogeli*	692	OM948897
	No access to a yard, a street or a forest							
	Lives with other cats							
	No history of infestation ectoparasites and uses ectoparasiticide							
Female, 1 year old, neutered	Lives in house	Prostration	Leukocytosis	Positive	Negative	*Babesia vogeli*	744	OM948898
	No access to a yard, a street or a forest	Vomiting	Neutrophilia					
	Lives with other cats	FeLV positive	Monocytosis					
	No history of infestation ectoparasites and uses ectoparasiticide							
Female, 12 years old, neutered	Lives in house	Apathy	Thrombocytopenia	Positive	Negative	*Babesia vogeli*	662	ON227489
	No access to a yard, a street or a forest	with neoplasm						
	Lives with other cats							
	History of flea infestation and doesn’t use ectoparasiticide							
Female, 4 years old, neutered	Lives in house	Urine with blood	Leukopenia	Positive	Negative	*Babesia vogeli*	615	OM948900
	No access to a yard, a street or a forest		Neutropenia					
	Lives with other cats		Thrombocytopenia					
	No history of infestation ectoparasites and doesn’t use ectoparasiticide							
Male, 1 year old, neutered	Lives in apartment	No clinical changes	Eosinophilia	Positive	Negative	*Babesia vogeli*	351	OM948901
	No access to a yard, a street or a forest		Neutrophilia					
	Lives with other cats		Lymphopenia					
	No history of infestation ectoparasites and doesn’t use ectoparasiticide							
Male, 2 years old, neutered	Lives in apartment	No clinical changes	Thrombocytopenia	Positive	Negative	*Babesia vogeli*	614	OM948902
	No access to a yard, a street or a forest							
	Lives with other cats							
	Has history of flea infestation and doesn’t use ectoparasiticide							
Male, 1 year old, un-neutered	Lives in house	No clinical changes	Erythrocytosis Lymphocytosis	Negative	Positive	*Cytauxzoon* sp.	241	OM948903
	Has access to a yard, a street and a forest							
	Lives with other cats							
	Has history of flea infestation and uses ectoparasiticide							

## Data Availability

Not applicable.

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
