# Peer review of "Piroplasm Infection in Domestic Cats in the Mountainous Region of Rio de Janeiro, Brazil"

_pathogens, 2022, doi:10.3390/pathogens11080900_

Round 1
Reviewer 1 Report
The authors present a regional molecular study to detect piroplasm infection in cats from the city of Teresópolis, Rio de Janeiro, Brazil.
This study is pertinent because it provides new information on feline piropasmosis. Currently, knowledge of this infection in cats is still scarce worldwide, as there are many epidemiological and clinical aspects about the feline piropaslmosis that are not well known. Therefore, and studies of this types are needed to fill this information gap.
In this study, 250 cat samples were examined for piroplasm DNA using molecular tools. The authors also provide epidemiological and clinical data on all included cats. The results of this study contribute to improve the current knowledge about feline piroplasmosis in Brazil.
The manuscript is generally well written. However, the following aspects of the presentation should be improved:
- The introduction is very short. A little more information on the epidemiology and clinical picture of feline piroplasmosis worldwide and specifically in Brazil should be included.
- The material and methods section should be presented before than the results section.
In addition, this paper could ultimately be accepted for publication with the following modifications (I attach the manuscript pdf with the parts of interest highlighted) :
Title: I think that it is more appropriate to write the title as “Piroplasms infection in domestic cats…”
Abstract. Line 36. Here you write that “nucleotide sequences of Babesia vogeli were identified in six cats, while Cytauxzoon sp. was identified in one cat” But in the results that you show (Table 5), there is one cat positive to Rangelia vitalii and five (not six) positive to B. vogeli. Like the same in Figure 1. There are only five sequences to B. vogeli and not 6. Please clarify it.
Results.
Line 72. Where it is written “most of the cat owners, 74.8% 71 (187/250), reported they lived in houses”. And the 25.2% rest of the cats, where did they live? Could you add this information too?
Line 80. Could you also include the percentage of cats with a history of tick infestation?
Line 88. Please, add the Confidence Interval (CI) of the total positive rate obtained. (i.e., 95% CI: ).
Table 1. Why do your certain times use the Chi-squared test and others Fisher’s Exact test? Could you explain it in methods (statistical analysis section)?
Line 120. After the sentence “The only clinical and hematological changes evidenced in the positive cats were those 120 described in Table 3”. Please, write the total number of cats that showed at least one clinical sign. Did all positive cats show at least one clinical sign or were any asymptomatic?
Table 3. Please write “Erytrogram” as red blood cell count and “Leukogram” as leukocyte count
Table 4. It is the same as table 2. The title does not correspond with the table itself.
Line 138. In the following sentence “Of the seven samples positive for piroplasms detected by molecular analysis in the 137 blood of cats in the, six” Please, delete “in the” before “six”.
In addition, where you wrote “six were discovered by Nested-PCR”, but were not there seven? Please clarify it. Why did you not include a Phylogenetic tree with the positive sequence from Rangelia vitalii found?
Line 139. Where is it written “all six showed nucleotide” Do you mean “all five”? Because there are five animals that are positive Babesia vogeli.
Title of table 5. Please delete the comma after “General”
Line 230 (Title of Figure 3). Please write “thigh” as coxae
Discussion
Line 239. Before “Such higher” I think that there is one more space
Line 251. Where you stated, “majority of positive cats did not have access to the backyard”. Does it mean that these cats were strictly indoor cats?
Lines 254-255. Could you add the information about the origin of cats? How many cats out of the 250 were adopted or rescued? Or at least if any of the seven piroplasm-positive cats were adopted?
Line 260. Do you have information on other vector borne diseases screened in the sick cat population?
Lines 270-271. Could you provide any possible explanation why positive samples were not amplified for other targets?
Lines 273-275. But also, it would be interesting to analyze the similarities with dog isolates of B vogeli form the same area. That, it turns out, is not unusual, as in endemic areas for canine babesiosis, several canine piroplasms species have been found to infect cats, using molecular tools.
Materials and methods
Line 352. Could you add the amount of blood that was taken (expressed as mL)?
Line 358. Could you add what Giemsa dilution you used to stain the blood smears?
Line 364. Please write global leukometry as leukocyte count
Lines 386-387. In the sentence “In addition, Cytauxzoon sp. DNA was analyzed using the protocol described”. Add after “described by Birkenheuer (2006)”
Line 401. Please, specify which type of positive controls you used. For example, positive sample of Babesia canis
Lines 432-433. The sentence “The identification of the ticks is presented at the lowest possible taxonomic level based on morphological analyses” should be incorporated into the taxonomic identification of tick section of the methodology.
Lines 433-434. The following sentences can be deleted: “The result of the infestation was descriptively associated with piroplasm infection. In addition, this paper describes the results of the search for piroplasms in the arthropods”. Because only one tick was found.
Conclusion
Line 441. Why didn't you also include Rangelia vitalii in the conlcusion?

Author Response
Dear reviewer 1
We would like to thank you for the suggested corrections
All were performed in the text and are marked in green. The corrections made were:
More information on the epidemiology and clinic of piroplasmosis in the world and mainly in Brazil was included in the introduction with other studies.
The position of the material item and methods follows the model of the journal.
The title has been changed as suggested.
We would like to apologize because table 5 had the wrong result, the same has already been corrected for Babesia vogeli.
The housing of the other Lines 91 – 93 cats was included
Tick ​​infestation history Lines 125 – 126 was included
The sampling confidence interval 466-471 was included
Chi-square was performed to analyze variables with more than two categories and Fisher's Exact for analysis of variables with two categories. Lines 561 - 562
The number of positive cats and the clinical changes was described in Lines 174 – 176
Fixed for “red blood cell count” and “leukocyte count” In table 3
Sorry, the correct table 4 has been inserted.
It was removed “in the” line 194
Lines 196-204 words were inserted to make it clear that two different PCR protocols were performed with different targets
Lines 198-199 has been corrected information.
Title of table 5, the information “General” was removed
Line 292 was inserted the word coxae
Line 301 Deleted space.
Line 314. According to the owner's report, most cats did not have access to the yard. This information has been highlighted in the text.
We did not include information regarding the origin of the animals in the questionnaire. But by conversations during the clinical examination we verified that the positive cats were rescued. (Line 318 – 320).
Research has not been carried out for other vector-borne parasites in animals, as far as we know.
Lines 345 – 348, the discussion on non-amplification of DNA was inserted.
Line 367 – 372 was inserted the discussion of similarity about B. vogeli previously identified in dogs from the same region.
The amount of blood collected was entered (Line 464)
The Giemsa dilution has been entered (line 477-478)
Fixed to leukocyte count (line 484)
Described by Birkenheuer (2006) was inserted line 509
The control used was specified in line 535
The sentence “The identification of the ticks...analyses” was inserted in the methodology section” 497 – 499.
Deleted sentence “The result of the infestation was descriptively associated with piroplasm infection. In addition, this paper describes the results of the search for piroplasms in the arthropods”.
Thanks, The authors.
Reviewer 2 Report
Dear author,
The manuscript entitled: “Piroplasms in domestic cats in the mountainous region of Rio de Janeiro, Brazil” provides interesting new information about the piroplasm infection in cats. I read the manuscript with great interest. The aim of this study is interesting and the manuscript is well documented. It could be accepted after some major revisions, especially results and discussion should be more considered and improved. Nevertheless, overall it is a small study which adds only little to existing knowledge.
Some key epidemiological considerations are lacking e.g. study design, statistics, limitations etc. The prevalence of piroplasms present in this study seems to be underestimated considering the number of cats tested and the limited area from which these cats came.
Materials and Mehods
Lines 338-353: What were the conditions for the animals to be included in the study? Were all cats presented at the veterinary clinic included in the study? However, it is about only 250 cats.
Line 384: „gene fragments encoding GAPDH” is uses for detecting the presence of PCR inhibitors in DNA samples.
Line 393: The authors should mentioned which are „the adaptations” added to the NaCl extraction technique.
Results
Lines 71-87: A table with all this data would be much more useful and easier to follow Lines 218-219: “highlighting the thrombocytopenia revealed by PCR in cats positive for B. vogeli”. - This sentence should be corrected.
Discussion
Lines 272-273: “five positive samples showed nucleotide sequences of Babesia vogeli”. In the results, the authors mentioned that B. vogeli was identified in 6 samples “all six showed nucleotide 139 fragments with 99% to 100% identity with Babesia vogeli”
Thank you!
Author Response
Dear reviewer 2
We would like to thank all the suggestions made. These are marked in the text in yellow. In addition, corrections to the text are commented on in this letter.
The study design, statistics and limitations were inserted in the text.
In lines 444 – 445, 466 – 471 (it has green also requested by reviewer 1), 428 – 434.
The inclusion of animals. It is important to emphasize that the Veterinary Clinic belongs to a private college in the region. In this way, the consultations take longer so that the students can learn. In addition, the collections were carried out during the pandemic period, when the number of cases in Brazil was high, which generated a restriction in the number of appointments at the clinic. Even so, we managed to reach the minimum sample size to stay within the expected confidence interval, taking into account the sample size and the frequency of other studies carried out in Brazil.
The objective of performing PCR with GAPDH line 506 has been corrected
Line 527-528 the adaptation of the technique was inserted
We agree that tables make some data easier to understand, but the article already has five tables. In this way, we tried to place in the tables the results referring to parasitology, the main objective of the article.
Lines 278 – 279 the sentence has been changed.
Line 349 the number was corrected
Thanks for the suggestions.
Round 2
Reviewer 2 Report
The manuscript could be accepted in present form.
Thank you!